# Using Cryo-ET to distinguish platelets during pre-acute myeloid leukemia from steady state hematopoiesis

Yuewei Wang[1,2,12], Tong Huo[1,12], Yu-Jung Tseng [3,4,12], Lan Dang[1], Zhili Yu[1], Wenjuan Yu[1,5], Zachary Foulks[6,7], Rebecca L. Murdaugh[3,8], Steven J. Ludtke [1,9], Daisuke Nakada [3,4,8,10✉] & Zhao Wang [1,9,11✉]

Early diagnosis of acute myeloid leukemia (AML) in the pre-leukemic stage remains a clinical challenge, as pre-leukemic patients show no symptoms, lacking any known morphological or numerical abnormalities in blood cells. Here, we demonstrate that platelets with structurally abnormal mitochondria emerge at the pre-leukemic phase of AML, preceding detectable changes in blood cell counts or detection of leukemic blasts in blood. We visualized frozen-hydrated platelets from mice at different time points during AML development in situ using electron cryo-tomography (cryo-ET) and identified intracellular organelles through an unbiased semi-automatic process followed by quantitative measurement. A large proportion of platelets exhibited changes in the overall shape and depletion of organelles in AML. Notably, 23% of platelets in pre-leukemic cells exhibit abnormal, round mitochondria with unfolded cristae, accompanied by a significant drop in ATP levels and altered expression of metabolism-related gene signatures. Our study demonstrates that detectable structural changes in pre-leukemic platelets may serve as a biomarker for the early diagnosis of AML.

[1] Verna and Marrs McLean Department of Biochemistry and Molecular Biology, Baylor College of Medicine, Houston, TX, USA. [2] Department of Vascular Surgery, the Affiliated Hospital of Qingdao University, Qingdao, China. [3] Department of Molecular and Human Genetics, Baylor College of Medicine, Houston, TX, USA. [4] Graduate Program in Translational Biology and Molecular Medicine, Baylor College of Medicine, Houston, TX, USA. [5] Department of Pathology, the Affiliated Hospital of Qingdao University, Qingdao, China. [6] Department of Chemistry, Missouri University of Science and Technology, Rolla, MO, USA. [7] The summer undergraduate research program (SMART program), Baylor College of Medicine, Houston, TX, USA. [8] Graduate Program in Developmental Biology, Baylor College of Medicine, Houston, TX, USA. [9] CryoEM/ET core, Baylor College of Medicine, Houston, TX, USA. [10] Dan L Duncan Comprehensive Cancer Center, Baylor College of Medicine, Houston, TX, USA. [11] Department of Molecular and Cellular Biology, Baylor College of Medicine, Houston, TX, USA . [13] These authors contributed equally: Yuewei Wang, Tong Huo, Yu-Jung Tseng. ✉email: nakada@bcm.edu; zhaow@bcm.edu

Acute myeloid leukemia (AML) is a heterogeneous malignancy characterized by defective hematopoietic differentiation, leading to the accumulation of immature leukemic blasts and suppression of normal hematopoiesis[1]. AML is the most common type of leukemia diagnosed in adults and accounts for the highest percentage of leukemic deaths[2]. AML develops quickly with only a 2-month median overall survival (OS) for untreated patients[3], emphasizing the need for early diagnosis. For younger patients, the OS and complete remission (CR) rate decrease dramatically when the time from diagnosis to treatment (TDT) is more than 5 days[4]. To improve the prognosis of AML patients, early diagnosis and immediate initiation of therapy are critical to reduce morbidity and mortality. Generally, the diagnosis of AML and specific subtypes are based on blood tests, bone marrow tests, and chromosome testing[5]. Diagnosis of the early stage of AML (defined as pre-AML) can be challenging because early in development AML is often asymptomatic, with no changes in blood tests, and no immature platelets detected in the bloodstream. By studying the early stages of AML, it may be possible to develop new diagnostics capable of early detection.

In AML, genetic lesions in hematopoietic stem/progenitor cells (HSPCs) produce developmental defects in cell lineages arising from them, including megakaryocytes that produce platelets[6,7]. Previous studies have shown that hemostasis disorders and bleeding tendency in AML patients could be due to not only reduced platelet counts, but also functional platelet defects[8–11]. Some AML patients exhibited decreased platelet size and expression of the activated membrane receptor (GPIIb/IIIa and GPIb), which led to reduced in vivo platelet activation[12,13]. Conventional transmission electron microscopy (TEM) was applied in structural studies of platelets from AML patients, which discovered decreased numbers of α-granules and an absence of dense granules[13].

However, the current understanding of platelet ultrastructure is derived mainly from TEM studies with conventional chemical fixation known to cause structural artifacts in platelet organelles and macromolecules[14–16]. Unlike conventional EM, vitrification can arrest cellular dynamics in intact platelets within milliseconds, thereby providing a physiological "snapshot" of whole platelets without artifacts produced by chemical fixation[17–19]. Facilitated by the development of electron cryo-tomography (cryo-ET), ultrastructural alteration of platelets has been found to be associated with diseases such as ovarian cancer[17]. In this study, we performed an ultrastructural analysis of platelets in an AML mouse model[20,21] at different time points after inducing AML. We characterized morphological changes and quantitively measured subcellular organelles within platelets providing observations of early-stage AML.

## Results

Since translocation between the mixed lineage leukemia (MLL) and AF9 genes (producing MLL-AF9) is often found in AML, we used a murine AML model driven by MLL-AF9 oncogene in our study, which is widely used to recapitulate AML in human[22,23]. Five populations of platelets were visualized in this study, drawn from un-irradiated wild-type (WT) mice, irradiated mice transplanted with MLL-AF9-transformed HSPCs at two time points (1-week: pre-AML and 3-week: AML), and irradiated mice with normal bone BMT at the same two time points(1-week control and 3-week control) (Fig. 1). The un-irradiated WT platelets served as a baseline for detecting induced changes, and the irradiated mice with normal BMT served as an additional control against radiation-induced as opposed to AML-induced changes. Historically, full annotation and analysis of a single platelet using this technique required a full week of human effort, dramatically

limiting the N (number of platelets) in such studies. Using newly developed automation techniques, we substantially reduced the time for analysis of each platelet and have a significant N for each population. Despite improvements, N in such studies is still rate and financially limited.

**Visualization and quantification of platelets from un-irradiated WT mice.** PRP was obtained from the blood of WT ($n = 3$) mice (designated as un-irradiated WT platelets hereafter), and a total of 94 platelet tilt series were collected and automatically reconstructed using EMAN2[24]. Reconstructed tomograms showed all un-irradiated WT platelets were in a quiescent state appearing as a discoid with a small number of pseudopods (Fig. 1, Supplementary Fig. 1 and Supplementary Movie 1)[25].

The platelet plasma membrane was relatively smooth, and the surface-connected open canalicular system (OCS) was found randomly dispersed on the otherwise featureless platelet exterior. Platelets and organelle membranes were clearly resolvable, indicating that we achieved spatial discrimination at nm resolution. Microfilaments in the cytoplasm could be identified in thinner regions of the tomograms, such as those near the plasma membrane. A coiled circumferential marginal band of microtubules is clearly observed. The cytoplasm of platelets was rich in glycogen and granules were randomly distributed in the cytoplasm. α granules with a single membrane had high morphological variability and the peripheral zone had less contrast than the central zone. The most distinguishing feature of dense granules was the electron-opaque spherical body within the organelle, separated from the enclosing membrane by an empty space producing a "bull's-eye" appearance[26]. Gamma granules with irregular dark densities were less common than α and dense granules. Mitochondria contained a double membrane feature, which was readily visible in all tomograms. The folded inner membrane cristae of mitochondria are readily identified.

In addition, 75.5% (71/94) of platelets had a tubule-like structure (TS) (Supplementary Fig. 1). TS has been observed in most platelets of Wistar Furth rats[27], giant platelets[28], and Medich's giant cell disease[29] using TEM. This structure was previously named as membranous inclusions[27] or scroll-like structure[29,30] because the membranes are flat sheets wrapped around each other once or several times forming a scroll. We named this feature TS because we found the membranes were repeatedly folding to form a tubule-like structure. This subcellular feature is clearly resolved in the tomograms, with a wide range of diameter and direction in cells. In contrast to previous studies, we found TS contained various types of organelles including α granules, dense granules, gamma granules, mitochondria, or glycogen. The function of TS is not yet clear: it may be the membrane storage to meet the maximum abduction of platelet activation, accumulation of glycogen particles to store energy or a channel for quick movement of organelles in platelets.

All sub-cellular features were processed in the same way and displayed to show a panorama of a whole platelet cell (Fig. 1). The accuracy of the segmentation was validated by visual inspection of each cell (Supplementary Fig. 1). To quantitatively compare and analyze features in platelets, we measured areas of whole platelets, mitochondria, α granules, and dense granules with a manual measurement tool in EMAN2 (see methods).

**AML accompanies multiple morphological changes in platelets.** To determine how AML affects the morphology of platelets, we transplanted either MLL-AF9-transformed HSPCs or normal bone marrow cells into irradiated recipient mice to induce AML or to reconstitute normal hematopoiesis, respectively[21,31]. AML mice exhibited increased numbers of white blood cells (WBC),

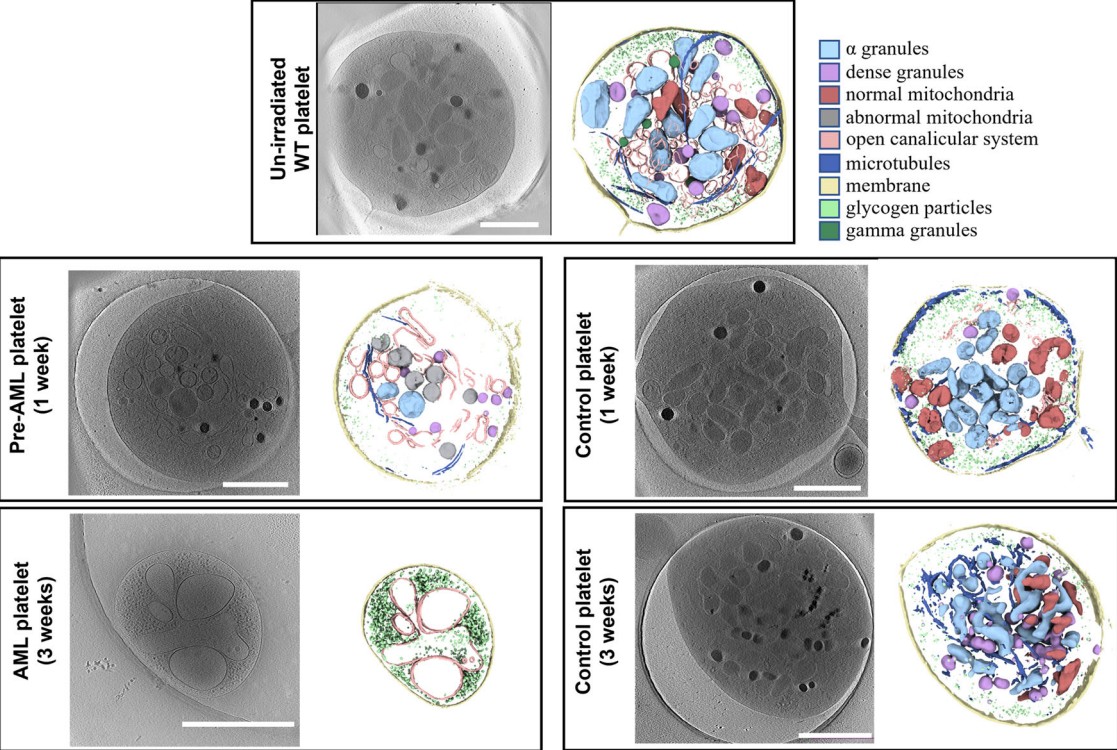

**Fig. 1 Cryo-ET of platelets from mice in different states.** Platelets from un-irradiated WT mice and BMT control mice had typical features including α granules, dense granules, gamma granules, mitochondria, intact membrane, microtubules, glycogen particles, and an open canalicular system (OCS). The platelets from AML mice (3 weeks) showed small round or oval shape without pseudopods. There were only glycogen particles and OCS inside. Platelets from pre-AML mice (1 week) can contain a mixture of normal and abnormal mitochondria. The example shown has only abnormal mitochondria. Color scheme used in segmentation is indicated in the panel. (Scale bar 1 μm).

decreased red blood cells (RBC) and platelets 3 weeks after transplantation. In contrast, mice with normal bone marrow cells exhibited normal numbers of blood cells (Table 1).

We collected data for 120 platelets from the recipient mice at 3 weeks post-transplantation and processed the data using the pipeline developed as described above. We found that 91.7% (110/120) of AML platelets contain only glycogen and OCS with roughly 21% of the normal cell size ($15.6 \pm 1.00$ vs. $75.6 \pm 1.84$ $10^5$ nm², $P < 0.001$) and empty vacuole vesicles or absent vesicles, called 'ghost platelets' (Fig. 2, Supplementary Fig. 2 and Supplementary Movie 2). The remaining 8.3% (10/120) of platelets exhibited similar shape and size as normal platelets, and intracellular features could be identified clearly including α granules, dense granules, gamma granules, mitochondria, the plasma membrane, circumferentially coiled microtubule, glycogen particles, and open OCS (same feature as shown in Fig. 1). Activated platelets are easily detected by the presence of pseudopods and microtubular changes. While our conditions were designed to avoid activation, occasional activated platelets were observed in the control and un-irradiated populations, but not in the AML population. Platelets from recipient mice transplanted with normal bone marrow cells exhibited platelets of similar size to normal platelets ($68.4 \pm 3.03$ vs. $75.6 \pm 1.84$ $10^5$ nm², $P = 0.099$), and no other structural abnormalities were detected (Fig. 1 and Supplementary Table 1). These results demonstrate that platelets in mice with frank AML are replaced with structurally abnormal platelets.

**Cryo-ET of platelets from pre-AML mice.** We then tested whether structurally abnormal platelets were detectable in mice at the early stage of AML. To this end, we collected PRP from mice ($n = 3$) 1 week after transplanting MLL-AF9-induced AML cells

and analyzed them with cryo-ET. Recipient mice at this time point exhibited reduced blood cell counts due to the effects of sub-lethal irradiation, but showed improvement at 3 weeks[32,33]. In contrast to mice 3 weeks after transplantation, which exhibit leukocytosis and severe anemia, mice at 1-week post-transplantation did not show leukocytosis or anemia according to the blood count (Table 1). We thus defined this time point as pre-AML.

We imaged 72 platelets from pre-AML mice. We found that all platelets contain the subcellular features observed in platelets from un-irradiated WT mice, including α granules, dense granules, gamma granules, mitochondria, the plasma membrane, microtubule, glycogen particles, OCS, and TS (Fig. 1, Supplementary Fig. 2 and Supplementary Movie 3).

However, platelets from pre-AML mice showed increased average area of α granules (mean area of all α granules in the same platelet) ($1.6 \pm 0.08$ vs. $1.3 \pm 0.05$ $10^5$ nm², $P = 0.004$) and decreased number of α granules ($11.3 \pm 0.9$ vs. $15.9 \pm 0.8$, $P < 0.001$) compared to un-irradiated WT mice (Supplementary Fig. 3 and Supplementary Table 2). In addition, the number of dense granules in pre-AML platelets was reduced ($9.7 \pm 0.7$ vs. $12.5 \pm 0.8$, $P = 0.009$), so was the average area ($0.31 \pm 0.01$ vs. $0.34 \pm 0.01$ $10^5$ nm², $P = 0.029$) (Supplementary Fig. 3). There is no significant change on the number of mitochondria, while the average area of mitochondria increased in pre-AML platelets (Fig. 2b, Supplementary Fig. 3c, Supplementary Data 1).

To investigate whether the aforementioned statistical changes are caused by AML development or BMT, we imaged the platelets from control mice that underwent the same transplantation procedure. Applying the same statistical analysis revealed no significant change between BMT control and pre-AML group on the average area of α granules, number of α granules, average area

**Table 1 CBC tests for mice at different time points.**

| | Leukemic group (3 mice) | | | Control group (3 mice) | | | Normal range |
|---|---|---|---|---|---|---|---|
| | Before BMT | 1 week after BMT | 3 weeks after BMT | Before BMT | 1 week after BMT | 3 weeks after BMT | |
| WBC ($10^3$/mm$^3$) | 5.8/6.0/8.4 | 0.9/0.3/0.4 | 58.9/72.5/80.4 | 11.9/14.8/11.1 | 1.1/1.9/1.6 | 4.4/3.6/3.7 | 3.0–15.0 |
| RBC ($10^6$/mm$^3$) | 6.96/10.79/10.01 | 6.16/9.20/6.03 | 1.39/2.08/3.05 | 6.10/8.36/9.18 | 2.63/11.47/8.33 | 6.23/5.84/6.06 | 5.0–12.0 |
| PLT ($10^3$/mm$^3$) | 562/316/427 | 114/150/300 | 112/73/107 | 531/481/675 | 251/191/142 | 242/242/405 | 140–600 |

*CBC complete blood count, BMT bone marrow transplantation; Leukemic group, transplanted with leukemia progenitor and stem cells; Control group, transplanted with normal bone marrow cells, WBC white blood cell, RBC red blood cell, PLT platelet.*

**Fig. 2 Statistics of platelets from mice in different states. a** There were no differences in the size of platelets from pre-AML, control mice and un-irradiated mice, while the size of platelet from AML mice was significantly decreased. **b** The average area (AA) of mitochondria in pre-AML platelets increased compared with un-irradiated and control groups ($P < 0.05$). Data are presented as mean ± s.e.m. Student's $t$ test were used for statistical analyses. *$P \leq 0.05$.

of dense granules, number of dense granules, and number of mitochondria, except for the average area of mitochondria (Fig. 2b and Supplementary Fig. 3). The difference of α and dense granules between un-irradiated WT mice and pre-AML mice would be probably attributed to transplantation rather than the AML development. The number of mitochondria in each platelet spans a large range, from 1 to 23 per cell, and does not seem to be correlated with AML. However, the average area of mitochondria in pre-AML platelets is larger than that in both BMT control and un-irradiated platelets, which indicates the mitochondrial structure may suffer subtle changes during the development of AML.

**Emergence of abnormal mitochondrion precedes hematological changes in pre-AML.** The majority of the mitochondria of platelets in pre-AML mice were similar to platelets isolated from un-irradiated mice and BMT control mice, exhibiting irregular shapes and containing typical structural features like highly folded cristae and matrix layers. However, a significant subset (23.6% (17/72)) of pre-AML platelets had clearly abnormal mitochondria with a round shape and less reduced cristae and matrix (Fig. 3a, b).

The average area was slightly increased compared with un-irradiated mice ($0.7 \pm 0.03$ vs. $0.6 \pm 0.01$ $10^5$ nm$^2$, $P = 0.023$) and BMT control mice ($0.7 \pm 0.03$ vs. $0.5 \pm 0.04$ $10^5$ nm$^2$, $P = 0.011$) (Fig. 2b). The number of abnormal mitochondria (ratio of

abnormal/normal mitochondria in one cell) varied among platelets. Roughly 3/4 of platelets that contained abnormal mitochondria mostly contained only one abnormal mitochondrion, the remaining quarter had two to seven abnormal mitochondria per platelet (Fig. 3b). To better define and distinguish the morphological difference between normal and abnormal mitochondria, we use the term "circularity" to describe the roundness of individual mitochondrion. We developed a new EMAN2 tool to report the circularity of each mitochondrion by measuring perimeter and area (see Methods). The average circularity of abnormal and normal mitochondria is $0.9681 \pm 0.0036$ and $0.7288 \pm 0.0219$, respectively (Fig. 3c, Supplementary Data 2). It is noteworthy that the shape of abnormal mitochondria is close to a perfect circle, and is very consistent among all abnormal mitochondria, while the circularity of normal mitochondria is lower and the deviation is larger due to its native morphological heterogeneity.

Key to its possible use as a diagnostic, not even a single instance of an abnormal mitochondria was observed in either un-irradiated mice or mice transplanted with normal bone marrow cells at the same time point as pre-AML mice (Fig. 1). The lack of abnormal mitochondria in the irradiated control group eliminates the possibility that this was a transitory effect due to irradiation.

In addition to the morphology of mitochondria, the surrounding features in pre-AML platelets were also altered. To investigate the changes in the environment surrounding mitochondria in an unbiased and uniform way, we isolated a 1 μm$^2$ surrounding each

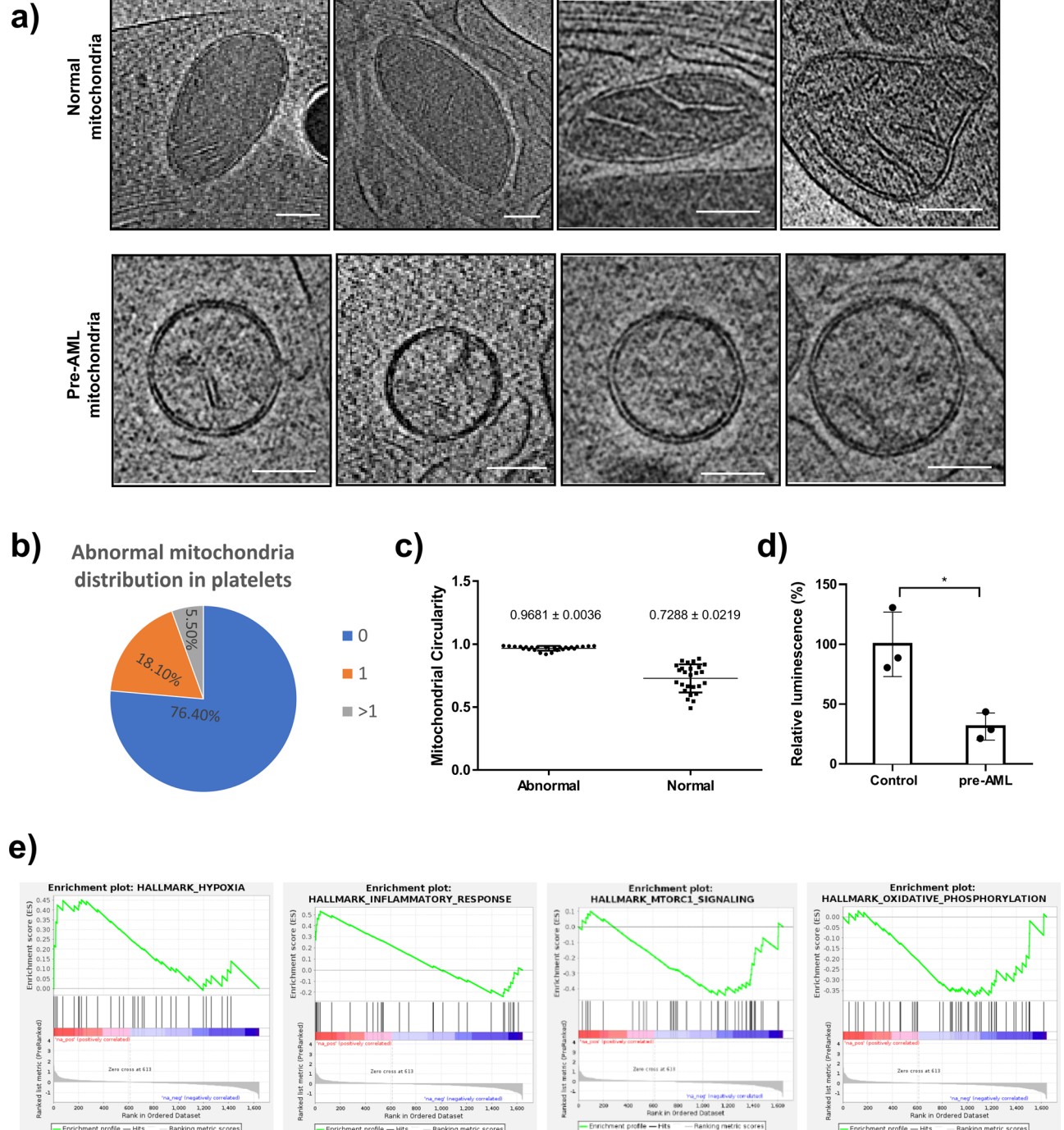

**Fig. 3 Abnormal mitochondria in pre-AML platelets. a** Roughly a quarter of mitochondria in pre-AML platelets were abnormal with reduced cristae and matrix (scale bar 100 nm). **b** 18.1% of the pre-AML platelets had one abnormal mitochondrion and 5.5% have more than one abnormal mitochondrion. **c** The statistical analysis of mitochondrial circularity ($P < 0.0001$) **d** ATP measurement assay for mitochondria in control and pre-AML platelets. **e** Gene set enrichment analysis (GSEA) on the platelets from pre-AML and control mice. "HYPOXIA" and "INFLAMMATORY_RESPONSE" were up-regulated and "MTORC1_SIGNALING" and "OXIDATIVE_PHOSPHORYLATION" were down-regulated in pre-AML platelets, compared with control platelets. Data in (**c**) and (**d**) are presented as mean ± s.e.m. Student's t test was used for statistical analyses. *$P \leq 0.05$.

mitochondrion (Supplementary Fig. 4a). Interestingly, abnormal mitochondria are all surrounded by single-layer spherical vesicles (Supplementary Fig. 4). Given that the interior of these vesicles have the same contrast as the cytosol when viewed through a central slice, the vesicles are likely to have a similar content to cytosol. As a result, these empty vesicles appear as spheres due to uniform osmosis pressure. These vesicles range from 100 to

300 nm in diameter and are often found in clusters. Like the abnormal mitochondria, these clustered vesicles were found only in pre-AML platelets. Mitochondria provide energy for platelet activation, aggregation, secretion of procoagulant molecules[34,35], and also are responsible for aging and the intrinsic apoptosis that regulates the lifespan of platelets[36,37]. The abnormal mitochondria in the pre-AML platelets observed in our study

could be the sign or consequence of its dysfunction or abnormal degradation, which would further trigger apoptosis of platelets. Given the concurrent appearance of empty spherical vesicles, it implies that mitochondria are undergoing degradation or mitophagy and spherical vesicles are possibly formed by re-conjugation of the broken membrane fractions due to nonspecific hydrophobic interactions. In specific tomograms, connections between empty vesicles and mitochondria are captured by chance, resembling a general vesicle budding process (Supplementary Fig. 4). These round vesicles show similar contrast and structural features as OCS in the AML platelets (Fig. 1). The observation of abnormal mitochondria in the pre-AML platelet may indicate a shorter lifespan and potential apoptosis of platelets, which may become the smaller sized 'ghost platelets'.

**Platelet mitochondria functional change at early stage of AML.** To acquire a better understanding of the mitochondrial changes in pre-AML platelets, we performed an ATP assay to measure the platelet intercellular ATP level and evaluate how ATP production is affected by abnormal mitochondria. Platelets from control and pre-AML mice were extracted, followed by a luciferase-based ATP assay. ATP production in pre-AML platelets exhibited a large reduction compared with BMT control platelets (Fig. 3d, Supplementary Data 3). In addition, we also conducted RNA-seq for platelets collected from BMT control and pre-AML mice. Consistent with the observed structural alterations, GSEA analysis revealed an upregulation in inflammatory response and hypoxia as well as a downregulation in genes involving in oxidative phosphorylation in pre-AML platelets, which is consistent with the model that the mitochondrial function is impaired in the platelets from pre-AML mice (Fig. 3e).

## Discussion

Based on cryo-ET imaging, platelets can be classified into three distinct stages based on the phase of AML development: (1) Healthy platelets from un-irradiated WT mice that have a normal size ($\sim75.6$ $10^5 nm^2$) containing all subcellular features including α granules, dense granules, gamma granules, mitochondrion, the plasma membrane, microtubules, glycogen particles, OCS, and TS; (2) pre-AML platelets with a similar size ($\sim73.0$ $10^5$ $nm^2$) as un-irradiated WT and BMT control platelets but with a fraction of mitochondria ($\sim25\%$) that have a round shape and diluted matrix; and (3) AML platelets, in which the majority appear as "ghost platelets" with signs of apoptosis, are five times smaller than BMT control platelets ($\sim15.6$ $10^5 nm^2$), and only contain glycogen storage and an open canicular system with empty vacuole vesicles.

It has been known that impaired proliferation or maturation of megakaryocytes (MKs) in bone marrow is a major cause of abnormal platelets such as thrombocytopenia[38,39]. The formation of platelet-specific organelles, proteins, membrane systems, and other contents occurs after and thus is closely related to MK maturation. Therefore, the structural change of mitochondria found in pre-AML mice in our study could be caused by defects in MK maturation. Given that the lifespan of platelets in circulation is 5 days in mice[40–42], virtually all of the platelets in circulation at the time point 1 week after BMT originated from abnormal MKs. These platelets could represent the changes of MKs and be a marker for pre-AML.

Mitochondria, referred as the "powerhouse of the cell", play critical roles in various essential biological processes including proliferation, differentiation, and metabolism[43]. There have been intensive studies on platelet mitochondria, demonstrating that the mitochondria impairment is associated with platelet dysfunction

and diseases, such as sepsis, Alzheimer's disease (AD) and Parkinson's disease (PD)[44–46]. In our study, by taking advantage of cryo-ET, we are able to detect the ultrastructural alteration of mitochondria in platelets in pre-AML phase. More importantly, the function of mitochondria was also impaired, with cellular ATP level decreased by more than 50%. GSEA analysis reveals downregulation in genes related to mitochondria function such as oxidative phosphorylation. Since the rate of ATP turnover and oxidative phosphorylation is tightly regulated by mitochondria during platelet's activation to meet the energy demand, the incapability of platelet in AML would be likely attributed to the dysfunction of mitochondria[47]. In pre-AML, mitochondria with unfolded inner membranes did not appear different in density than the surrounding cytosol, suggesting that swelling and dilation of the inner membrane compartment caused matrix dilution or efflux of matrix components. Taken together, we demonstrate that both the structure and function of mitochondria exhibit early signs of impairment in the pre-AML phase. Mitochondrial apoptosis and/or mitophagy may lead to cell shrinkage and shedding of platelet microparticles[48,49]. We speculate that the abnormal mitochondria that lacks spherical cristae could be the product of apoptosis or mitophagy and the small empty platelets we found in fully developed-AML phase are the outcome after platelet apoptosis.

The progressive depletion of normal platelets and accumulation of abnormal platelets parallels the development of AML from the pre-AML stage without detectable changes in blood cell counts to frank AML. Our data suggest that the emergence of abnormal mitochondria and surrounding empty vesicles could be an early sign of AML initiation. These features can be detected in raw cryo-ET images, but our semi-automatic reconstruction pipeline allows clear distinction and quantification of different compartments and changes in the mitochondrion (Supplementary Fig. 4b). However, even without additional developments, Cryo-ET itself can easily detect mitochondrial changes and the coordinated appearance of empty vesicles. Our observation of clearly visible changes in a large fraction of platelets at early stages of AML, at which other diagnostics fail, presents an opportunity for a new class of non-invasive diagnostics capable of early detection and early treatment of this often-fatal disease. It may also be possible to develop more straightforward laboratory tests to detect these abnormalities and produce a diagnostic inexpensive enough to become routine.

Since the feature of "abnormal mitochondria" appears exclusively in the pre-leukemia platelet population, it can be used to predict whether a subject has the malignancy status. From observed data of 72 preleukemic cells, the probability that a pre-leukemic cell exhibits the abnormal mitochondria phenotype is 23.6%. As a result, by testing 12 cells, we would have a 96% chance of detecting at least one abnormal cell.

It is possible that the observed mitochondrial changes are specific to this AML model system or to a subset of AML cases. Recapitulating these results during AML development in humans will be difficult, as we have no strategy to identify pre-AML patients, due to a lack of other diagnostics. With the relatively low prevalence of AML in the population, identifying candidates would require a prohibitively large study. While mice are appropriate models to study human platelets given their overall similarities, human and mice platelets do exhibit morphological differences, and murine platelets are smaller and more numerous and display much greater granule heterogeneity than human platelets[50,51]. Future work is needed to establish whether AML patients exhibit similar changes as presented here with the mice model.

## Methods

**Mice**. The mouse alleles used in this study were Ubc-GFP (C57BL/6-Tg(UBC-GFP)30Scha/J, JAX Stock #004353)[52] on a C57BL/6 background. CD45.1 (B6.SJL-Ptprca Pepcb/BoyJ, JAX Stock #002014) or C57BL/6 mice was used as transplant recipient mice. Female mice (8–12 weeks old) were used and housed in AAALAC-accredited, specific-pathogen-free animal care facilities at Baylor College of Medicine (BCM). All procedures were approved by the BCM Institutional Animal Care and Use Committee. The BCM Institutional Biosafety Committee approved all other experimental procedures.

**Bone marrow transplantation (BMT)**. MLL-AF9 retrovirus was prepared by transfecting the pMIG-MLL-AF9-IRES-GFP construct with pCL-ECO into HEK293T cells as previously described[31]. Hematopoietic stem and progenitor cells (Lin⁻c-kit⁺Sca-1⁺) were sorted and incubated in X-Vivo 15 (Lonza, Allendale, NJ) supplemented with 50 ng/ml SCF, 50 ng/ml TPO, 10 ng/ml IL-3, and 10 ng/ml IL-6 (all from Peprotech, Rocky Hill, NJ) for 24 h prior to retrovirus spin infection. For primary recipients, 50 000 cells were transplanted into lethally irradiated C57BL/6 mice (500 cGy, twice, with at least 3 h interval). Secondary transplantation was performed by transplanting 100 000 GFP + cells from primary recipient mice into sub-lethally irradiated mice (650 cGy). For the control, mice underwent transplantation of normal bone marrow cells.

**Cryo-ET of platelets**. Platelet-rich plasma (PRP) was separated from drawn blood samples and then vitrified for cryo-ET. The mice were in supine position after inhalation anesthesia by diethyl ether. The abdomen and thoracic cave were open to expose heart. Then, 100 μL whole blood was obtained by puncturing right atrium and put in 500 μL EP tube with 3.8% sodium citrate (SC) as the anticoagulant (blood vs SC 9:1). The whole blood is centrifuged at 500 g to collect PRP. PRP (2–4 μL) mixed with 10 nm fiducial gold marker was applied to the glow-discharged Quantifoil grid and then blotted with calcium-free blotting paper using a Vitrobot (Mark IV, FEI Corp) and immediately plunged into liquid ethane at liquid nitrogen temperature to vitrify the platelets (1–3 blot, 1–3 s wait). PRP purification and cryo-preservation were maintained at room temperature. All platelet samples were vitrified for cryo-ET study within 2 h after blood draw.

The frozen grids with platelets were then transferred into one of three electron microscopes (JEM 3200FSC, FEI Polara F3 and Titan Krios) equipped with K2 Summit camera for imaging. Low-dose conditions were used to preserve structural integrity. Using SerialEM[53], tilt series were recorded at /300 keV on a Gatan 4096 × 4096 pixels K2 Summit direct election camera in counting mode with 0.2 s per frames exposure in (2, 4 and 8 sec). Single-tilt image series (±50°, 2° increment) were collected at a defocus range of 15–20 μm and a magnification range of 3000–4 000×, with a pixel size of 12.06 Å. The total electron dose per tomogram was 90–100 electrons/Å², as typical for cellular cryo-ET. Each tilt series has 51 tilt images with a fixed 2-degree increment.

**Platelet tomogram processing and quantification**. Image stacks were automatically aligned and reconstructed using e2tomogram in EMAN2[54], yielding 3D reconstructed tomograms (voxel 4k × 4k × 1k, ~17–19 Å per pixel sampling). Gold fiducials were present but not used explicitly, as the software uses any high contrast features for alignment. To enhance the contrast, tomograms were averaged by two (bin2) and filtered uniformly using a low-pass filter (set up at 33 Å) to reduce the noise and visualized in UCSF Chimera[24,55]. Structural features such as α granules, dense granules, gamma granules, mitochondria, microtubules, membrane systems, glycogen particles, and tubule-like system (TS) that was formed by membrane repeated folding were identified as previous study[26], segmented, and manually annotated using EMAN2, UCSF Chimera[55], and UCSF ChimeraX (Supplementary Fig. 1a)[56].

Semi-automated segmentation used a neural network based method[24] requiring user-provided training annotation on representative areas. We manually selected and labeled 10 regions containing obvious mitochondria as positive controls and 100 regions without mitochondria as negative controls in the same tomograms from different slices and orientation, respectively (Supplementary Fig. 1b). The trained network was then applied to the complete set of tomograms, reducing over a man-year of potential effort to a few weeks. Separate neural networks were trained for each type of organelle. The segmentation results were confirmed by human visualization at the end.

Structural features (α granule number, α granule area, dense granule number, dense granule area, mitochondria number, mitochondria area) were manually quantified for each platelet tomogram (Supplementary Fig. 1a). Though segmented and labeled in tomograms, microtubules were not assessed due to the low resolvability in Z direction which leads to uncertainty in determining clear boundaries of microtubules in some orientations. Low resolvability in the Z direction of the 3D tomogram reconstruction is mainly due to the missing wedge in cryo-ET data. The central slice of the 3D tomogram of the whole platelet was projected. The area of each platelet was measured in a single projection image with the assumption that the platelets have a uniform thickness (Supplementary Fig. 1c). Mitochondria area is expressed as the summed area of the central slice through each mitochondrion in a platelet divided by the area of the platelet in its central slice. α and dense granule areas were measured in the same way. The circularity of

mitochondria was assessed via the formula $C = 4 \times \pi \times A/P^2$, where C is the circularity, A is the area and P is the perimeter. For a circle, C = 1. All the quantification was performed by the same researcher to maintain consistency throughout the entire study.

The representative tomogram of platelet from un-irradiated, BMT 1-week control, BMT 3-week control, pre-AML and AML mice were deposited to EMDB (EMD-24287, EMD-24291, EMD-24298, EMD-24289 and EMD-24288, respectively).

**Predictive model and power analysis**. In our study, un-irradiated WT and control cells never exhibited abnormal mitochondria. We performed a post-hoc power analysis to put an upper bound on the probability of observing such false positives. With an N of 106 non-malignant cells and $P = 0.05$, we can state that false positive rate can be no more than 2.8%.

**RNA-seq**. Whole blood from control and pre-AML mice were collected and PRP was separated via Lymphoprep (Stemcell Technologies). Platelets were further purified from the plasma by staining with CD45 and Ter119 antibodies then depleting the CD45 or Ter119 positive cells through autoMACS (Miltenyi Biotec). Purified platelets were resuspended in Trizol and RNA purified according to the manufacturer's instructions. DNase I-treated RNA samples were purified using the QIAGEN MinElute kit before cDNA was made and amplified with a SMART-Seq2 protocol. The cDNA was then fragmented and barcoded for sequencing using a Nextera XT kit (Illumina, San Diego, CA). RNA-seq libraries were sequenced on an Illumina Sequencer (Illumina, San Diego, CA). Reads were mapped to mm10 using STAR (version 2.5.2b)[57], which was followed by differential expression analysis using DESeq2 (version 1.12.4)[58]. The PCA plot was generated with deep Tools.

**ATP measurement assay**. Platelets were collected from control or AML mice and resuspend in PBS. Platelet ATP level was measured using CellTiter-Glo® 2.0 Cell Viability Assay (Promega). Luminescence was determined by Infinite M200 PRO from Tecan.

**Statistical analysis of quantified platelet tomograms**. Experimental data were analyzed by SSPS 16.0 for Windows (SPSS Inc., Chicago, IL, USA). Data are presented as the means ± standard error of the mean (SEM). Differences between groups were assessed using the Student's t test or Mann–Whitney U-test and statistical significance was taken at $P < 0.05$.

**Statistics and reproducibility**. The platelet preparation, Cryo-ET imaging, ATP measurement and RNA-seq were all done from three independent mice each time under same treatment. For subcellular feature measurement and analysis, data were presented as means ± SEM. The Student's t test or Mann–Whitney U-test was used to assess differences. P values equal to or less than 0.05 were considered statistically.

**Reporting summary**. Further information on research design is available in the Nature Research Reporting Summary linked to this article.

## Data availability

The authors declare that the data supporting the findings of this study are available from the corresponding author W.Z. upon request. Source data related to the graphs and charts in this paper are provided in Supplementary Data 1–3. Tomographic reconstruction data that support the findings of this study have been deposited in EMDB with the accession codes EMD-24287, EMD-24291, EMD-24298, EMD-24289 and EMD-24288. The RNA-seq data have been deposited to GEO with the accession number GSE189844.

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

## Acknowledgements

We thank Margaret Goodell, Guizhen Fan, Issac Forrester, and Xinzhe Yu for suggestions and comments on the experiment and manuscript. This work is supported by Robert Welch Foundation (Q-1967-20180324), NIH-GMS (R01GM143380) and BCM BMB department seed funds to Z.W., NIH (R01GM080139 and P01GM121203) to S.L., NIH (F31DK112542) to R.L.M., and NIH (R01CA193235 and R01CA255813) to D.N. D.N. is a scholar of the Leukemia and Lymphoma Society. We acknowledge the use of the Cryo-EM supported by Advanced Technology CPRIT Cryo-EM/ET Core (1RP190602) at BCM and cryo-EM core at UTHealth Center at Houston.

## Author contributions

S.L., D.N., and Z.W. conceived and supervised the project. Y.W., T.H., and W.Y. prepared the platelet sample. T.H., Y.W., and Z.Y. performed data collection, processing, and analysis. Y.T. performed the treatment of the mice and CBC test. L.D. assisted in the utilization of EMAN2. Z.F. made the movies of tomograms. R.L.M. performed the RNA sequencing analysis. Y.W., T.H., and Z.W. wrote the manuscript with other authors' input.

## Competing interests

The authors declare no competing interests.
