## [Peer Review File · Communications Biology]

Reviewers' comments:

Reviewer #1 (Remarks to the Author):

Wang et al. use cryo-ET in order to perform early diagnosis of AML. Their authors show major changes in platelet organization that are detected at relatively low resolution. This is an exciting finding that may have consequences on the early detection of this severe disease. The findings are interesting and may impact the community.

Major points:

1. Although the authors used cryo-ET, it is not clear why cryo-ET is needed to distinguish membrane organelles. In fact, high-pressure freezing, freeze substitution, followed by sectioning (or cut-and-view block FIB approach) would allow to detect the granule alterations as well as the changes in mitochondria, at much higher contrast and maybe even higher resolution. Moreover, larger statistics (one section or block image would show a large number of cells) and lower costs would be experienced. Glembotsky et al. used conventional chemical fixation, therefore the quality of their images was rather low. In summary, this reviewer does not see the advantage of using cryo-ET. The power of cryo-ET is compromised by the sample thickness, thus in toto high resolution analysis is not possible and cannot be applied to human platelets (x1.5-2 larger).
2. The authors claim a resolution of 5 nm based on "visualizing membranes", unfortunately this criterion is not justified because the membranes are elongated along the axes of the beam (and membranes are phosphate rich structure and therefore highly scattering material). Indeed, the authors can only detect actin filaments (8 nm) in thinner cellular parts (P. 5) indicating that the resolution of these tomograms is much lower than 5 nm. With a maximal resolution of 3.4-3.8 nm and first zero at 4.4-4.9 nm, it is unlikely to have 5 nm resolution, without a proper CTF correction and averaging.

Other points:

1. The title indicated that cryo-ET can distinguish... Please modify as a method can only be used to distinguish.
2. P.6. The authors filtered their tomogram to Ny frequency ($1/(2 \times 17-19 \text{ \AA})$). This should not be considered as a low pass filter.
3. The raw data sets should be uploaded to an open data base (e.g. EMPIAR) and the extension number should be indicated.

Reviewer #2 (Remarks to the Author):

In this manuscript, Wang et al. report that a proportion of platelets in the pre-leukemic phase of AML exhibit structurally abnormal mitochondria. They further propose that these changes in structural changes in pre-leukemic platelets detected by cryo-electron tomography could be a biomarker for the early diagnosis of AML.

Overall, the topic is relevant and the results are interesting. While the claims made by the authors may be of interest, there are some complementary experiments that should be performed before further considerations.

- Authors mentioned that recapitulating these results in human will be difficult since it implies that the analysis will be done on AML patients at a later diagnosis stage. It is true however it is an important point and it will be worth to already investigate whether the platelets of AML patients display the same abnormalities than the platelets found in the AML mouse model. Another option to analyse platelet structure during human AML development is to perform PDX in mice.
- In Figure 1B, in the AML platelet there is no mitochondria. Is it always the case? If yes could the authors precise whether there is or not mitochondria. Moreover, why are the mitochondria gone? Are they degraded? via mitophagy? or are they excluded outside the platelets? Authors mentioned the potential implication of mitophagy in pre-AML platelets, therefore it would be interesting to investigate why and what are the mechanisms explaining the loss of mitochondria in AML platelets.

and the structural changes in pre-AML platelets.

- In pre-AML platelets, mitochondria shape and structure are different compared to normal platelet. What are the consequences on mitochondria functions? are they altered? OCR, ATP production etc...should be analysed.

- Is the ER structure affected in pre-AML and in AML platelets? Authors have to analyse the ER structure upon AML development and its potential interaction with the mitochondria.

Reviewer #3 (Remarks to the Author):

This interesting paper aims to demonstrate that substantial structural changes can be detected at an early stage by Cryo-ET in pre-leukemic platelets in mice who are otherwise asymptomatic and with no alteration in blood tests. Whilst normally platelets play a critical role in haemostasis, in cancer patients platelets are known to be hyperactive and promote tumour growth and metastasis through multiple pathways. Cancer-related hyperactivity of platelets is thought to be associated with changes in platelets ultrastructure, such as cytoskeletal rearrangement and organelles' alteration, but the visualization of these changes has been fraught due to the limitations imposed by methods that use plastic embedding and chemical fixation. In the last 10 years, key advances in Cryo-EM software and hardware have made it possible to accurately image the ultrastructure of entire cells using Cryo-ET. The relatively small sizes of platelets are particularly conducive to whole cellular cryo-ET.

By using whole cellular Cryo-ET together with semi-automatic image analysis for the recognition and quantification of number and size of organelles, the authors suggest that drastic changes occur in the overall shape and depletion of organelles in platelets of murine models of AML when compared to healthy mice. Furthermore, and more interestingly the authors conclude that more subtle but detectable changes in pre-leukemic murine platelets can be observed and suggests that detection of these changes by Cryo-ET could be used to confirm early AML diagnosis.

These findings will be of interest to structural biologists developing cryo-ET protocols to characterize the 3D ultrastructure of intact cells at high resolution. Equally, the results will be of interest to haematologists and cancer cell biologists with interest in platelet biology looking to understand the dynamics of this cell and its implication with disease. The manuscript is therefore of potential interest to the wide-readership of Communications Biology and publication is recommended subject to the authors addressing the comments and suggested corrections below.

General comments:

1. The main weakness of the paper is that the main conclusion that pre-AML platelets have alteration in organelles' size and shape is not strongly supported by the data. As described in the specific comments below, comparisons with the appropriate normal BMT controls are missing, stringent definition of normal v abnormal mitochondria are lacking and some of the Cryo-ET images presented are either not of high quality or fail to convince of what is being purportedly shown. Could the authors please address the points made in the "specific comments" section below to try and reinforce the conclusions.

2. The authors fail to discuss the granules findings in light of the fact that mouse platelet granules differ from those of human platelets. Specifically mouse alpha granules are more heterogeneous in shape and size and are fewer per platelet section than human alpha granules. This heterogeneity in mouse and the difference in numbers, might be big enough to render the findings of this study not applicable to human platelet granules. The authors should address this criticism.

3. The authors fail to compare their results with those of a similar Cryo-ET study of ovarian cancer platelets. In this study similar quantitative measures showed significant difference between control and cancer platelets for microtubule length, the number of mitochondria, and the percentage of the total platelet area occupied by mitochondria. The microtubule length is not reported in this manuscript, whilst the number of mitochondria and the percentage of the mitochondrial area compared to the total platelet area are unchanged. The authors need to present their results for the microtubules and should address the discrepancy in findings between this paper and the ovarian cancer platelet paper.

4. The authors have not performed predictive modelling to convince the reader that the Cryo-ET quantification methodology used here is robust enough to predict the status of a platelet (healthy,

pre-AML, AML). The authors should perform predictive modelling and assess whether the sample size in this study is sufficient to provide the appropriate predictive power.

Specific Comments

- a) Page 5 Cryo-ET of platelets. More details are needed in the Methods section PRP preparation: How were the platelets derived? Specifically, how was blood collected to avoid platelet activation (a caution that is particularly applicable to mice)? Details such as anaesthesia and blood withdrawal procedures should be given. Also provide protocol for the isolation of PRP from blood, temperature of storage of blood and PRP samples prior to application on the grid. EM grid preparation: What type of grids were used? Were they glow discharged?
- b) Page 6 Line: 117 When identifying various structural features, how are these defined? Also, TS, which is discussed in the Result section, is not defined in the Methods section.
- c) Page 6 line 116: correct 33A into 33Å
- d) Page 9 line 176: it is not clear which activated platelets the authors refer to? The platelets discussed in the text are quiescent (page 8, line 159) with small number of pseudopodia not activated presumably.
- e) Page 10 Line 186: when describing TS introduce a reference to Figure S2A.
- f) Page 12 line 222: use decimal points consistently
- g) Page 13 line 224: "...exhibited uniform shapes and dimensions..." please clarify "uniform" and give dimensions.
- h) Page 13 line 230: use decimal points consistently. The two measurements appear to be in the wrong order.
- i) Page 13 line 240: In order to demonstrate that pre-AML platelets contain the subcellular features observed in healthy platelets (WT and normal BMT after one week), comparative statistics for lambda granules, microtubules, glycogen particles, OCS and TS should be presented alongside those for alpha granules, dense granules and mitochondria.
- j) Page 14 line 244: the size of the platelet of normal BMT mice after one week should be presented as the appropriate control here, in Fig2E and in Table S1.
- k) Page 14 Figure 2, Table S2 and Page 15 lines 258-266: The numbers, areas and area percentages for the various organelles (alpha granules, dense granules and mitochondria) should be shown also for the appropriate control mice (normal BMT after one week).
- l) Page 15 Line 271: "...abnormal, round mitochondria..." how are normal and abnormal defined and divided into subsets? In Figure 3A top left image appears similar to second image on the right in terms of shape. This is important because most platelets containing "abnormal" mitochondria only contained one abnormal mitochondrion per platelet. Also, the images shown in Figure S3B in relation to the mitochondria in control platelet (1 week) are not clear nor convincing and better images should be provided.
- m) Page 17 Line 278: The average area increase is very small and should be compared to the area for normal BMT after one week, not to the area of normal mice.
- n) Page 17 Line 291 and Figure S3C: the 1 micron square box centred around the mitochondria should be shown in Figure S3C.
- o) Figure S3C does not appropriate support the claims. In the left "mito-surrounding in pre-AML platelet" image, the mitochondria indicated by the white arrow does not appear evidently abnormal (it is not rounded and spherical in shape and has some evident invaginations) and it does not appear connected with or in close proximity to the indicated empty vesicle. Conversely in the right "mito-surrounding in healthy platelet" image, the mitochondria indicated by the white arrow appears in close proximity to what appears to be an empty vesicle. No images are shown for the appropriate control (normal BMT after one week).
- p) Page 17 Line 292: "...abnormal mitochondria are all closely associated with spherical vesicles" this claim should be substantiated by distances.
- q) Page 18 Line 302: the concurrent appearance of empty vesicles and abnormal mitochondria, together with the criticism of the data above, does not necessarily imply that the pre-AML platelets' mitochondria are undergoing degradation or mitophagy. To demonstrate that these vesicles are of mitochondrial origin more evidence would be needed such as for example a positive stain for LC3. Otherwise the authors should suggests that alternative explanations are possible.

Response to Reviewers

We thank the reviewers for their overall positive comments and insightful suggestions to our manuscript. Our point-to-point responses to Reviewers' comments and modified manuscript (highlighted in blue in the revised manuscript) are provided below.

Reviewers' comments:

Reviewer #1 (Remarks to the Author):

Wang et al. use cryo-ET in order to perform early diagnosis of AML. Their authors show major changes in platelet organization that are detected at relatively low resolution. This is an exciting finding that may have consequences on the early detection of this severe disease. The findings are interesting and may impact the community.

We are encouraged by Reviewer 1's positive comments that "This is an exciting finding that may have consequences on the early detection of this severe disease." and "The findings are interesting and may impact the community."

Major points:

1. Although the authors used cryo-ET, it is not clear why cryo-ET is needed to distinguish membrane organelles. In fact, high-pressure freezing, freeze substitution, followed by sectioning (or cut-and-view block FIB approach) would allow to detect the granule alterations as well as the changes in mitochondria, at much higher contrast and maybe even higher resolution. Moreover, larger statistics (one section or block image would show a large number of cells) and lower costs would be experienced. Glembotsky et al. used conventional chemical fixation, therefore the quality of their images was rather low. In summary, this reviewer does not see the advantage of using cryo-ET. The power of cryo-ET is compromised by the sample thickness, thus in toto high resolution analysis is not possible and cannot be applied to human platelets (x1.5-2 larger).

As a new technique, cryoET can image frozen, hydrated platelets, both from mouse and human, and preserve native subcellular morphology with sufficient resolution of fine structural details (Wang et al., PNAS, 2015). To clarify, as our main conclusion is to directly measure native organelles in platelets, traditional EM or FIB which causes structural artifacts will not be suitable for this study.

2. The authors claim a resolution of 5 nm based on "visualizing membranes", unfortunately this criterion is not justified because the membranes are elongated along the axes of the beam (and membranes are phosphate-rich structures and therefore highly scattering material). Indeed, the authors can only detect actin filaments (8 nm) in thinner cellular parts (P. 5) indicating that the resolution of these tomograms is much lower than 5 nm. With a maximal resolution of 3.4-3.8 nm and first zero at 4.4-4.9 nm, it is unlikely to have 5 nm resolution, without a proper CTF correction and averaging.

We have clarified our statement about resolvability. This statement has been modified by observing nanometer resolution features of organelles and we did not use high-resolution in the

later analysis. See line 212. *“Platelets and organelle membranes were clearly resolvable, indicating that we achieved spatial discrimination at nm resolution...”*

Other points:

1. The title indicated that cryo-ET can distinguishes... Please modify as a method can only be used to distinguish.

We have modified the title as *“Using Cryo-ET to distinguish platelets during pre-acute myeloid leukemia from steady state hematopoiesis”*

2. P.6. The authors filtered their tomogram to Nq frequency ($1/(2 \times 17-19 \text{ \AA})$). This should not considered as a low pass filter.

This is indeed a misunderstanding due to the ambiguity of our original text. Please see a modified description of low-pass filter applied at 30% of the Nq frequency at 12Å resolution in revised text at line 118. *“To enhance the contrast, tomograms were averaged by two (bin2) and filtered uniformly using a low-pass filter (set up at 33 Å) to reduce the noise...”*

3. The row data sets should be uploaded to an open data base (e.g. EMPIAR) and the extension number should be indicated.

We uploaded the tomographic reconstructions to EMDataBank. The accession number has been added in the methods. *“The representative tomogram of platelet from un-irradiated, BMT 1-week control, BMT 3-week control, pre-AML and AML mice were deposited to EMDB (EMD-24287, EMD-24291, EMD-24298, EMD-24289 and EMD-24288, respectively).”*

Reviewer #2 (Remarks to the Author):

In this manuscript, Wang et al. report that a proportion of platelets in pre-leukemic phase of AML exhibit structurally abnormal mitochondria. They further propose that these changes in structural changes in pre-leukemic platelets detected by cryo-electron tomography could be a biomarker for the early diagnosis of AML.

Overall, the topic is relevant and the results are interesting. While the claims made by the authors may be of interest, there are some complementary experiments that should be performed before further considerations.

Reviewer 2 points out that “Overall, the topic is relevant and the results are interesting” and “these changes in structural changes in pre-leukemic platelets detected by cryo-electron tomography could be a biomarker for the early diagnosis of AML”. We thank this reviewer’s suggestions and revised the manuscript accordingly.

- Authors mentioned that recapitulating these results in human will be difficult since it implies that the analysis will be done on AML patients at later diagnosis stage. It is true however it is an important point

and it will be worth to already investigate whether the platelets of AML patients display the same abnormalities than the platelets found in the AML mouse model. Another option to analyse platelet structure during human AML development is to perform PDX in mice.

This is indeed a good future direction for our research raised by the reviewer. Unfortunately, there is no experimental data currently available to provide any plausible result from either AML patients or PDX mice. We have toned down the conclusion of the current study limited to our mice model. Please check the updated text at line 438. *“It is possible that the observed mitochondrial changes are specific to this AML model system or to a subset of AML cases. Recapitulating these results during AML development in humans will be difficult, as we have no strategy to identify pre-AML patients, due to a lack of other diagnostics. With the relatively low prevalence of AML in the population, identifying candidates would require a prohibitively large study. While mice are appropriate models to study human platelets given their overall similarities...”*

- In Figure 1B, in the AML platelet there is no mitochondria. Is it always the case? if yes could the authors precise whether there is or not mitochondria. Moreover, why are the mitochondria gone? Are they degraded? via mitophagy? or are they excluded outside the platelets? Authors mentioned the potential implication of mitophagy in pre-AML platelets, therefore it would be interesting to investigate why and what are the mechanisms explaining the loss of mitochondria in AML platelets and the structural changes in pre-AML platelets.

Thank you for these enlightening questions. 91.7% (110/120) of AML platelets contain only glycogen and OCS without any mitochondria, α granules, dense granules, and λ granules. The rest (10/120) of the AML platelets look normal with all the granules that can be visualized in the cell. Whether mitochondria and other organelles are degraded in AML via mitophagy or excluded outside the cell indeed are interesting mechanical questions, requiring further investigation in the future. We have modified the discussion and mentioned all these possible reasons, please see line 411. *“In pre-AML, mitochondria with unfolded inner membranes did not appear different in density than the surrounding cytosol, suggesting that swelling and dilation of the inner membrane compartment caused matrix dilution or efflux of matrix components. Taken together, we demonstrate that both the structure and function of mitochondria exhibit early signs of impairment in the pre-AML phase. Mitochondrial apoptosis and/or mitophagy may lead to cell shrinkage and shedding of platelet microparticles (ref. 53,54). We speculate that the abnormal mitochondria that lacks spherical cristae could be the product of apoptosis or mitophagy and the small empty platelets we found in fully developed-AML phase are the outcome after platelet apoptosis.”*

In addition, we also conducted functional assay for mitochondria in pre-AML platelets, which showed mitochondrial function was changed simultaneously when the structure was changed. Please see the reply below.

- In pre-AML platelets, mitochondria shape and structure are different compared to normal platelet. What are the consequences on mitochondria functions? are they altered? OCR, ATP production etc... should be analysed.

Thank you for your suggestion. We have performed an RNA-seq analysis and measured ATP levels in platelets isolated from pre-AML recipient mice. As shown in new Figure 3D and E, we found that pre-AML platelets had significantly lower ATP than platelets isolated from control bone marrow cells. We also found that pre-AML platelets had lower expression of genes involved in oxidative phosphorylation and the mTOR pathway, consistent with the notion that

pre-AML platelets are metabolically less active, while genes involved in hypoxia and inflammation are activated. Please see our updated text in line 368 and Figure 3D and 3E. "Platelet mitochondria functional change at early stage of AML..."

- Is the ER structure affected in pre-AML and in AML platelets? Authors have to analyse the ER structure upon AML development and its potential interaction with the mitochondria.

We did not find ER in this study. Platelets only have dense tubular system (DTS), which is a smooth endomembrane system originating from the endoplasmic reticulum of megakaryocytes (Shilpi, 2018). DTS is not clearly presented in every platelet and does not have a reliable feature for structural analysis due to its nature of membrane. (Harry, 2017; Josef, 2015)

Reviewer #3 (Remarks to the Author):

This interesting paper aims to demonstrate that substantial structural changes can be detected at an early stage by Cryo-ET in pre-leukemic platelets in mice who are otherwise asymptomatic and with no alteration in blood tests. Whilst normally platelets play a critical role in haemostasis, in cancer patients platelets are known to be hyperactive and promote tumour growth and metastasis through multiple pathways. Cancer-related hyperactivity of platelets is thought to be associated with changes in platelets ultrastructure, such as cytoskeletal rearrangement and organelles' alteration, but the visualization of these changes has been fraught due to the limitations imposed by methods that use plastic embedding and chemical fixation. In the last 10 years, key advances in Cryo-EM software and hardware have made it possible to accurately image the ultrastructure of entire cells using Cryo-ET. The relatively small sizes of platelets are particularly conducive to whole cellular cryo-ET.

By using whole cellular Cryo-ET together with semi-automatic image analysis for the recognition and quantification of number and size of organelles, the authors suggest that drastic changes occur in the overall shape and depletion of organelles in platelets of murine models of AML when compared to healthy mice. Furthermore, and more interestingly the authors conclude that more subtle but detectable changes in pre-leukemic murine platelets can be observed and suggests that detection of these changes by Cryo-ET could be used to confirm early AML diagnosis.

These findings will be of interest to structural biologists developing cryo-ET protocols to characterize the 3D ultrastructure of intact cells at high resolution. Equally, the results will be of interest to haematologists and cancer cell biologists with interest in platelet biology looking to understand the dynamics of this cell and its implication with disease. The manuscript is therefore of potential interest to the wide-readership of Communications Biology and publication is recommended subject to the authors addressing the comments and suggested corrections below.

We are encouraged by Reviewer 3's positive overall comments and recommend publication: "These findings will be of interest to structural biologists developing cryo-ET protocols to characterize the 3D ultrastructure of intact cells at high resolution.", "will be of interest to hematologists and cancer cell biologists with interest in platelet biology looking to understand the dynamics of this cell and its implication with disease."

General comments:

1. The main weakness of the paper is that the main conclusion that pre-AML platelets have alteration in organelles' size and shape is not strongly supported by the data. As described in the specific comments

below, comparisons with the appropriate normal BMT controls are missing, stringent definition of normal v abnormal mitochondria are lacking and some of the Cryo-ET images presented are either not of high quality or fail to convince of what is being purportedly shown. Could the authors please address the points made in the “specific comments” section below to try and reinforce the conclusions.

The main conclusion for pre-AML has been modified as suggested by the Reviewer 3. We also included data analysis of BMT controls and regenerated all the statistical analysis. The stringent definition (circularity) of abnormal and normal mitochondria in cryoET visualization has been added in the revised text (Figure 2,3 and S3).

2. The authors fail to discuss the granules findings in light of the fact that mouse platelet granules differ from those of human platelets. Specifically mouse alpha granules are more heterogeneous in shape and size and are fewer per platelet section than human alpha granules. This heterogeneity in mouse and the difference in numbers, might be big enough to render the findings of this study not applicable to human platelet granules. The authors should address this criticism.

We have clarified our statement about the heterogeneous in the mouse granules and add discussion in line 438 to point out this object feature may vary in human platelets. *“It is possible that the observed mitochondrial changes are specific to this AML model system or to a subset of AML cases. Recapitulating these results during AML development in humans will be difficult, as we have no strategy to identify pre-AML patients, due to a lack of other diagnostics...”*

3. The authors fail to compare their results with those of a similar Cryo-ET study of ovarian cancer platelets. In this study similar quantitative measures showed significant difference between control and cancer platelets for microtubule length, the number of mitochondria, and the percentage of the total platelet area occupied by mitochondria. The microtubule length is not reported in this manuscript, whilst the number of mitochondria and the percentage of the mitochondrial area compared to the total platelet area are unchanged. The authors need to present their results for the microtubules and should address the discrepancy in findings between this paper and the ovarian cancer platelet paper.

Though it is segmented and labeled in the tomogram, microtubes are often coiled together, and it is difficult to determine the boundaries in certain views. This makes it difficult to measure the length of the microtubes in the current study. To avoid possible errors caused by inaccurate measurements, we did not include microtubes in the measurement and further analysis.

We have added a clarification for not including microtubules in the statistical analysis (line 135). *“Though segmented and labeled in tomograms, microtubules were not assessed due to the low resolvability in Z direction which leads to uncertainty in determining clear boundaries of microtubules in some orientations. Low resolvability in the Z direction of the 3D tomogram reconstruction is mainly due to the missing wedge in cryo-ET data.”*

4. The authors have not performed predictive modelling to convince the reader that the Cryo-ET quantification methodology used here is robust enough to predict the status of a platelet (healthy, pre-AML, AML). The authors should perform predictive modelling and assess whether the sample size in this study is sufficient to provide the appropriate predictive power.

Thank you for your suggestion. We have performed a post-hoc power analysis and estimate false positive rate base on our observation. (Method: Line 151) *“In our study, un-irradiated WT and control cells never exhibited abnormal mitochondria. We performed a post-hoc power analysis to put an upper bound on the probability of observing such false positives. With an N of 106 non-malignant cells and P=0.05, we can state that false positive rate can be no more than 2.8%.”*

Specific Comments

a) Page 5 Cryo-ET of platelets. More details are needed in the Methods section PRP preparation: How were the platelets derived? Specifically, how was blood collected to avoid platelet activation (a caution that is particularly applicable to mice)? Details such as anaesthesia and blood withdrawal procedures should be given. Also provide protocol for the isolation of PRP from blood, temperature of storage of blood and PRP samples prior to application on the grid. EM grid preparation: What type of grids were used? Were they glow discharged?

We have added technical details in revised methods. Please see line 95. *“Platelet-rich plasma (PRP) was separated from drawn blood samples and then vitrified for cryo-ET. The mice were in supine position after inhalation anesthesia by diethyl ether. The abdomen and thoracic cage were open to expose heart. Then, 100 μ L whole blood was obtained by puncturing right atrium and put in 500 μ L EP tube with 3.8% sodium citrate (SC) as the anticoagulant (blood vs SC 9:1).”*

b) Page 6 Line: 117 When identifying various structural features, how are these defined? Also, TS, which is discussed in the Result section, is not defined in the Methods section.

We thank the reviewer for this point and have now included various structural features in the text (Line 121 and 216). *“...tubule-like system (TS) that was formed by membrane repeated folding were identified as previous study”, “ α granules with a single membrane had high morphological variability and the peripheral zone had less contrast than the central zone. The most distinguishing feature of dense granules was the electron-opaque spherical body within the organelle, separated from the enclosing membrane by an empty space producing a “bull’s-eye” appearance (ref. 28). Gamma granules with irregular dark densities were less common than α and dense granules. Mitochondria contained a double membrane feature, which was readily visible in all tomograms...”*

c) Page 6 line 116: correct 33A into 33Å

Thank you for your advice. We have revised it. (Line 119)

d) Page 9 line 176: it is not clear which activated platelets the authors refer to? The platelets discussed in the text are quiescent (page 8, line 159) with small number of pseudopodia not activated presumably.

Thank you for this point. We have removed this sentence since it did not relate to our conclusion in this study and may be a little confusing.

e) Page 10 Line 186: when describing TS introduce a reference to Figure S2A.

Thank you for the suggestion. We revised it and added “Figure S2A” at the end of the sentence. See line 224. *“In addition, 75.5% (71/94) of platelets had a tubule-like structure (TS) (Figure S2A). TS has been observed in most platelets of Wistar Furth rats (ref. 33), giant platelets (ref. 34), and Medich's giant cell disease (ref. 35) using TEM.”*

f) Page 12 line 222: use decimal points consistently

We revised it according to the suggestion, please see line 256. *“...the normal cell size (15.6 ± 1.00 vs. 75.6 ± 1.84 10^5 nm^2 , $P < 0.001$)”*. For better comparison and consistence with previous publication on platelet (Wang et. al , PNAS, 2015), we used the same unit and order of magnitude (10^5 nm^2) for the same measurement representational paradigm through our study.

g) Page 13 line 224: “...exhibited uniform shapes and dimensions...” please clarify “uniform” and give dimensions.

Thank you for the suggestion. 8.3% of platelets in AML were in a quiescent state and had a similar shape and size as normal platelets. The uniform means a similar shape and size. We revised this sentence. See line 258. *“The remaining 8.3% (10/120) of platelets exhibited similar shape and size as normal platelets....”*

h) Page 13 line 230: use decimal points consistently. The two measurements appear to be in the wrong order.

Please see the reply above for (f).

i) Page 13 line 240: In order to demonstrate that pre-AML platelets contain the subcellular features observed in healthy platelets (WT and normal BMT after one week), comparative statistics for lambda granules, microtubules, glycogen particles, OCS and TS should be presented alongside those for alpha granules, dense granules and mitochondria.

Thanks for the suggestion. We added the BMT control data to compare with our experimental data. Features of alpha granules, dense granules and mitochondria were analyzed and compared. Abnormal mitochondria only presenting in pre-AML platelets, which is our main conclusion in this study, is not changed. After combining the BMT control data, we changed some of our previous conclusion and made new figures and analysis based on the new data. Please see line 295. *“To investigate whether the aforementioned statistical changes are caused by AML development or BMT, we imaged the platelets from control mice that underwent the same transplantation procedure. Applying the same statistical analysis revealed no significant change between BMT control and pre-AML group on the average area of α granules, number of α granules, average area of dense granules, number of dense granules, and number of mitochondria, except for the average area of mitochondria (Figure 2B and S3). The difference of α and dense granules between un-irradiated WT mice and pre-AML mice would be probably attributed to transplantation rather than the AML development.”*

We did not include the statistic for gamma granules, OCS and TS because platelet not always contain these organelles. We did not include microtubules and glycogen in the quantitative analysis due to the similar reason mentioned in the reply to other reviewer. Glycogen may overlap and microtubules often coiled together. This makes it difficult to determine boundaries of these two organelles in certain views. To avoid possible error caused by inaccurate measurement, we did not include statistics analysis of these objects in the platelet. We have made explanation in our revision text. (Line 135) *“Though segmented and labeled in tomograms, microtubules were not assessed due to the low resolvability in Z direction which leads to uncertainty in determining clear boundaries of microtubules in some orientations. Low resolvability in the Z direction of the 3D tomogram reconstruction is mainly due to the missing wedge in cryo-ET data.”*

j) Page 14 line 244: the size of the platelet of normal BMT mice after one week should be presented as the appropriate control here, in Fig2E and in Table S1.

We re-made the Figure 2 and Table S1 and added BMT control data.

k) Page 14 Figure 2, Table S2 and Page 15 lines 258-266: The numbers, areas and area percentages for the various organelles (alpha granules, dense granules and mitochondria) should be shown also for the appropriate control mice (normal BMT after one week).

We added the statistical analysis data for control mice according to review's suggestion. Please see Figure 2 and Figure S3.

l) Page 15 Line 271: "...abnormal, round mitochondria..." how are normal and abnormal defined and divided into subsets? In Figure 3A top left image appears similar to second image on the right in terms of shape. This is important because most platelets containing "abnormal" mitochondria only contained one abnormal mitochondrion per platelet. Also, the images shown in Figure S3B in relation to the mitochondria in control platelet (1 week) are not clear nor convincing and better images should be provided.

The normal mitochondria are well defined as in previous studies, our structural features are consistent with previous cryoET studies (Wang et. al , PNAS, 2015). The abnormal mitochondria only observed in the current study, is defined as an organelle with double membranes that contain fewer cristae and matrix features than normal mitochondria. The abnormal mitochondria also present a round shape in contrast to an irregular shape in normal mitochondria. In the revision, we include a newly developed software tool to directly estimate mitochondria roundness and provide range of the roundness in the abnormal and normal mitochondria. See line 334. "We developed a new EMAN2 tool to report the circularity of each mitochondrion by measuring perimeter and area (see Methods). The average circularity of abnormal and normal mitochondria is 0.9681 ± 0.0036 and 0.7288 ± 0.0219 , respectively (Figure 3C). It is noteworthy that the shape of abnormal mitochondria is close to a perfect circle, and is very consistent among all abnormal mitochondria, while the circularity of normal mitochondria is lower, and the deviation is larger due to its native morphological heterogeneity."

We have also included an improved image to replace previous Figure S3B (Figure S4A now). Due to the data loss during the compression of the original file, the quality of the image would be diminished. We would submit the original version for the publication.

m) Page 17 Line 278: The average area increase is very small and should be compared to the area for normal BMT after one week, not to the area of normal mice.

A comparison between pre-AML and control mice has been added in the revised manuscript (Figure 2B).

n) Page 17 Line 291 and Figure S3C: the 1 micron square box centred around the mitochondria should be shown in Figure S3C.

We have re-made the Figure S4A (previous Figure S3C) to show the 1 micron area as reviewer suggested.

o) Figure S3C does not appropriately support the claims. In the left "mito-surrounding in pre-AML platelet" image, the mitochondria indicated by the white arrow does not appear evidently abnormal (it is not rounded and spherical in shape and has some evident invaginations) and it does not appear connected with or in close proximity to the indicated empty vesicle. Conversely in the right "mito-surrounding in healthy platelet" image, the mitochondria indicated by the white arrow appears in close proximity to what appears to be an empty vesicle. No images are shown for the appropriate control (normal BMT after one week).

We have re-made the Figure S4A (previous Figure S3C) and added 1-week BMT control data. As we mentioned above, we have included a quantitative roundness measurement in the mitochondria and demonstrate this directly shows difference between abnormal and normal mitochondria. Our ATP and RNA seq results also suggest the function of mitochondria is decreased in the pre-AML stage, further supporting our conclusion of the paper. As we mentioned in the methods, we can easily separate vesicles from mitochondria due to the double membrane feature and folded cristae and matrix layers by cryoET.

p) Page 17 Line 292: "...abnormal mitochondria are all closely associated with spherical vesicles" this claim should be substantiated by distances.

We have re-made the Figure S4A to show that the empty vesicles are always present within 1 micron area when abnormal mitochondria spotted.

q) Page 18 Line 302: the concurrent appearance of empty vesicles and abnormal mitochondria, together with the criticism of the data above, does not necessarily imply that the pre-AML platelets' mitochondria are undergoing degradation or mitophagy. To demonstrate that these vesicles are of mitochondrial origin more evidence would be needed such as for example a positive stain for LC3. Otherwise the authors should suggest that alternative explanations are possible.

The discussion now been revised as: *“Given the concurrent appearance of empty spherical vesicles, it implies that mitochondria are undergoing degradation or mitophagy and spherical vesicles are possibly formed by re-conjugation of the broken membrane fractions due to nonspecific hydrophobic interactions...”* (Line 359)

REVIEWERS' COMMENTS:

Reviewer #1 (Remarks to the Author):

The authors answered all my points and concerns.

Reviewer #2 (Remarks to the Author):

The authors have more or less responded effectively to reviewer's previous concerns.

Reviewer #3 (Remarks to the Author):

I feel that with the corrections and the introduction of additional data as requested the authors have considerably improved their manuscript, which I now consider suitable for publication in Communication Biology.